# Transformer Doctor: Diagnosing and Treating Vision Transformers

**Jiacong Hu**[1,4]**, Hao Chen**[1]**, Kejia Chen**[2]**, Yang Gao**[6]**,**
**Jingwen Ye**[3]**, Xingen Wang**[1,6]**, Mingli Song**[1,4,5]**, Zunlei Feng**[2,4,5*]

[1]College of Computer Science and Technology, Zhejiang University,
[2]School of Software Technology, Zhejiang University,
[3]Electrical and Computer Engineering, National University of Singapore,
[4]State Key Laboratory of Blockchain and Data Security, Zhejiang University,
[5]Hangzhou High-Tech Zone (Binjiang) Institute of Blockchain and Data Security,
[6]Bangsheng Technology Co., Ltd.

`{jiaconghu,hao_chen_,chenkejia,roygao}@zju.edu.cn,`
`jingweny@nus.edu.sg,{newroot,brooksong,zunleifeng}@zju.edu.cn`

## Abstract

Due to its powerful representational capabilities, Transformers have gradually become the mainstream model in the field of machine vision. However, the vast and complex parameters of Transformers impede researchers from gaining a deep understanding of their internal mechanisms, especially error mechanisms. Existing methods for interpreting Transformers mainly focus on understanding them from the perspectives of the importance of input tokens or internal modules, as well as the formation and meaning of features. In contrast, inspired by research on information integration mechanisms and conjunctive errors in the biological visual system, this paper conducts an in-depth exploration of the internal error mechanisms of Transformers. We first propose an information integration hypothesis for Transformers in the machine vision domain and provide substantial experimental evidence to support this hypothesis. This includes the dynamic integration of information among tokens and the static integration of information within tokens in Transformers, as well as the presence of conjunctive errors therein. Addressing these errors, we further propose heuristic dynamic integration constraint methods and rule-based static integration constraint methods to rectify errors and ultimately improve model performance. The entire methodology framework is termed as Transformer Doctor, designed for diagnosing and treating internal errors within transformers. Through a plethora of quantitative and qualitative experiments, it has been demonstrated that Transformer Doctor can effectively address internal errors in transformers, thereby enhancing model performance. For more information, please visit https://transformer-doctor.github.io/.

## 1 Introduction

In the field of machine vision, models based on Transformers [1, 2] have gradually replaced convolutional neural networks as the mainstream approach. Particularly in recent years, various visual architectures improved upon Transformers have emerged incessantly [3–6], continuously pushing the performance boundaries of visual tasks. However, the vast and complex parameters of Transformers hinder researchers from gaining a deep understanding of their internal mechanisms [7],

---

[*]Corresponding author.

38th Conference on Neural Information Processing Systems (NeurIPS 2024).

thereby increasing the risks of applying them in sensitive domains [8]. This has spurred a considerable amount of work aimed at investigating the interpretability of Transformers to enhance their transparency [9–11].

Existing research on the interpretability of Transformers in machine vision primarily focuses on aspects such as the importance of input tokens [7, 9, 10, 12–14], the significance of internal modules [15], the the evolution and formation of features [16, 17], and the meanings of intermediate represnetations [18–20]. While these studies have somewhat improved the transparency of Transformers, the internal decision-making processes, such as mechanisms leading to errors, still warrant more systematic investigation. This is crucial for enhancing the transparency of Transformers and further improving their performance.

In fact, unlike in machine vision, theoretical studies on error mechanisms in biological vision have become quite mature [21–26]. Specifically, in the perceptual process of the biological visual system, visual information such as the spatial position, shape, size, color, and texture of objects is processed and refined in the primary visual cortex [27–29]. Subsequently, these different visual cues are integrated at higher stages for final recognition [30, 31]. Numerous studies have demonstrated that errors in object recognition may arise not only from failures in feature extraction but also from incorrect integration of correctly extracted features at higher stages [21–25]. Errors resulting from the improper integration of features are termed conjunction errors [26]. Furthermore, some research indicates that providing effective stimuli or cues during the integration process can enhance the correctness of information integration [22]. Inspired by this, we are interested in investigating whether Transformers exhibit similar mechanisms of feature integration and conjunction errors as those observed in biological vision during recognition. If such errors exist, can they be corrected akin to the mechanisms observed in biological vision?

To address these questions, we first proposed the *information integration hypothesis*, inspired by the biological visual. This hypothesis posits that Transformers continuously process and refines various mixed information at the primary stage and integrate them at the higher stage. When incorrect information is integrated, i.e., conjunction error occur, it results in erroneous recognition outcomes. To validate this hypothesis, we conducted extensive experimental analyses of the computational process of Transformers and found empirical evidence supporting the hypothesis. Specifically, we discovered dynamic information integration among tokens in Transformer's Multi-Head Self-Attention (MHSA) component and static information integration within tokens in the Feed-Forward Network (FFN) component, along with the presence of conjunction errors. Furthermore, we elucidated the reasons behind both dynamic and static integration. Building upon this, we proposed heuristic dynamic integration constraints for inter-token information integration and rule-based static integration constraints for intra-token information integration, enabling the rectification of conjunction errors in Transformers. We coined the entire approach as "Transformer Doctor", where the process of identifying errors based on the information integration hypothesis is referred to as diagnosing the Transformer, and the process of applying the hypothesis to rectify errors is referred to as treating the Transformer. Finally, we conducted extensive quantitative and qualitative experiments on mainstream Vision Transformer architectures, thoroughly validating the effectiveness and applicability of Transformer Doctor.

The contributions of this paper can be summarized as follows:

- We propose Transformer Doctor, the first framework for diagnosing and treating Vision Transformers. This framework validates and utilizes the proposed Information Integration Hypothesis, which posits that Transformers process and encode various mixed information at primary stages and integrate it at higher stages. When information is not correctly integrated, i.e., conjunction error occur, it results in prediction failures.

- In diagnosing Transformers, we identify the mechanisms of inter-token information dynamic integration and intra-token information static integration within Transformers, along with the occurrence of conjunctive errors. This provides a novel perspective for understanding the internal mechanisms of Vision Transformers.

- In treating Transformers, we propose heuristic dynamic constraints for inter-token information integration and rule-driven static constraints for intra-token information integration. These constraints offer an interpretable solution for optimizing Vision Transformers without introducing additional parameters or computational overhead during inference.

- Extensive qualitative and quantitative experiments are conducted on mainstream Vision Transformers, validating the effectiveness and applicability of the Transformer Doctor.

## 2 Related Works

In methods aimed at interpreting or understanding Transformers, whether in the field of machine vision or natural language processing, the primary approaches focus on the importance of input tokens [7, 12, 13, 9, 14, 10], the significance of internal modules [15], the the evolution and formation of features [16, 17], and the meanings of intermediate features [18–20] to understand the internal mechanisms of Transformers. Specific methodologies can be categorized as feature-based, attention-based, gradient-based, propagation-based, perturbation-based, projection-based, or a combination of these approaches. For instance, in feature-based methods, the primary focus is on analyzing or statistically evaluating the intermediate features within Transformers [16] to understand the internal representation structure and feature distribution [17]. Attention-based methods mainly utilize the raw attention weights [7, 12, 13] or linear combinations of multi-layer attention weights [9] to compute the relative importance of input tokens. Gradient-based methods focus on computing gradients of attention weights [32, 33], intermediate features [11, 34], or inputs [14] to understand the differences in token importance. Propagation-based methods employ techniques like Layer-wise Relevance Propagation (LRP)[35, 36] for attribution analysis of input tokens[10, 37–39] or investigate the importance of heads in Transformers [15]. Perturbation-based methods involve perturbing inputs or features and measuring the impact on model performance [40–43] or Shapley value [14, 44]. Projection-based methods, such as linear probes [17, 45, 46], project intermediate representations into human-understandable spaces to comprehend the mechanism and significance of feature transformations in Transformers [18–20]. In contrast to the aforementioned research, this paper is inspired by studies on biological visual error mechanisms, aiming to explore whether Transformers exhibit similar mechanisms of information integration and connection errors as in biological vision, and how to rectify errors within Transformers.

In methods for improving Transformers, most approaches involve modifying the model architecture by introducing learnable parameters [3–6, 47] or enhancing data and features [48–52]. These improvement methods are mostly non-interpretable and are pre-defined before training, rather than targeting further enhancement of model performance from the perspective of diagnosing and treating internal error mechanisms based on existing models. Additionally, there are methods for improving models that do not require pre-definition before training but focus on non-Transformer or non-vision tasks, such as debugging and analyzing models in traditional machine learning [53–56], optimizing deep models [57, 58], and editing facts in natural language processing [59–62]. However, due to significant differences in architecture and tasks, these methods are not suitable for analyzing and correcting error mechanisms in Transformers to improve model performance.

In summary, this paper is the first work to investigate whether Transformers exhibit information integration and connection error mechanisms similar to biological vision, and the first work to explore how to further enhance model performance by diagnosing and treating internal errors in existing models.

## 3 Information Integration Hypothesis

In this section, we firstly propose the Information Integration Hypothesis for Transformers. Subsequently, we review the MHSA and FFN modules within the Transformer architecture, followed by an analysis of potential locations where the Information Integration Hypothesis may apply.

*Information Integration Hypothesis: Similar to biological vision, in machine vision, the Transformer continually processes and refines various mixed information in the primary stage, and integrates it in the advanced stage. When erroneous information is integrated, i.e., conjunction errors occur, it leads to incorrect predictions.*

### 3.1 Potential Information Integration in MHSA

When utilized for visual recognition tasks, a Transformer typically comprises $L$ blocks, each consisting of an MHSA module and an FFN module. The input to the block can be represented as

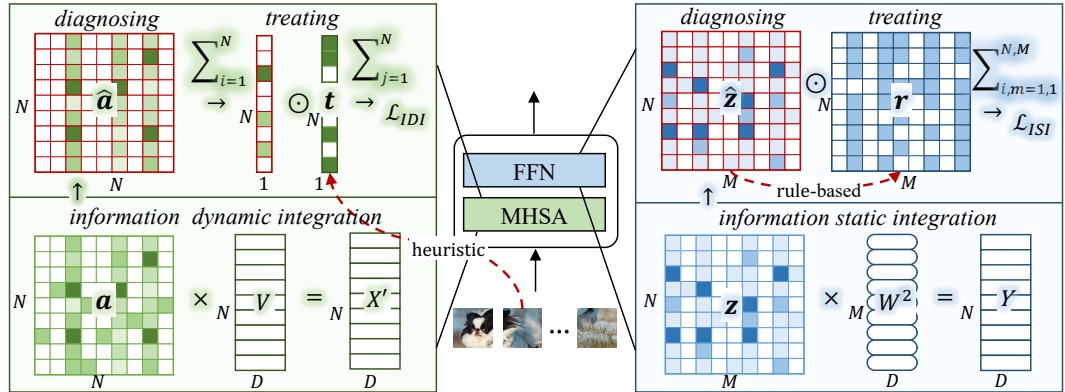

Figure 1: The methodology framework of Transformer Doctor. It begins by analyzing the dynamic integration of inter-token information in MHSA and the static integration of intra-token information in FFN, Subsequently, conjunction errors within them are diagnosed, and finally treated to enhance model performance.

$X \in \mathbb{R}^{N \times D}$, where $N$ and $D$ denote the number and dimensionality of tokens, respectively. The output $X' \in \mathbb{R}^{N \times D}$ of the MHSA module within this block can be computed as follows:

$$X' = \text{softmax}(\frac{1}{\sqrt{D}}QK^T)V, \tag{1}$$

where $Q = XW^{(Q)}, K = XW^{(K)}, V = XW^{(V)}$. $W^{(Q)} \in \mathbb{R}^{D \times D}$, $W^{(K)} \in \mathbb{R}^{D \times D}$, and $W^{(V)} \in \mathbb{R}^{D \times D}$ represent the parameter matrices for query, key, and value, respectively. Upon decomposing Eqn. (1), it can be observed that a certain token $X'_i \in \mathbb{R}^D$ in $X' \in \mathbb{R}^{N \times D}$ (e.g., the $i$-th token) is an integration weighted sum of all tokens in $V \in \mathbb{R}^{N \times D}$:

$$X'_i = \sum_{j=1}^{N} \mathbf{a}_{i,j} V_j, \quad \mathbf{a} = \text{softmax}(\frac{1}{\sqrt{D}}QK^T), \tag{2}$$

where $\mathbf{a} \in \mathbb{R}^{N \times N}$ represents the attention weights, which can also be referred to as integration weights within MHSA. It can be observed that each token $X'_i$ possesses its own integration weight $\mathbf{a}_i \in \mathbb{R}^N$. For simplicity, we have omitted the skip connections and bias terms in the above equation, and have considered only single-headed self-attention. Therefore, the concatenation and projection of multiple heads in MHSA are also omitted.

### 3.2 Potential Information Integration in FFN

Suppose the input to the FFN is $X' \in \mathbb{R}^{N \times D}$, then the output $Y \in \mathbb{R}^{N \times D}$ of the FFN can be represented as:

$$Y = \text{gelu}(X'W^{(1)})W^{(2)}, \tag{3}$$

where $W^{(1)} \in \mathbb{R}^{D \times M}$ and $W^{(2)} \in \mathbb{R}^{M \times D}$ represent the parameter matrices for the first and second linear layers, respectively. Similarly, for simplicity, we have omitted the skip connections and bias terms in the above equation. Comparing Eqn. (1) and Eqn. (3), it can be observed that apart from the activation function gelu in FFN and the softmax function in MHSA, as well as the constant $\frac{1}{\sqrt{D}}$, FFN also employs the query-key-value mechanism similar to MHSA. Motivated by this, we further decompose Eqn. (3) into a form similar to Eqn. (2), revealing that a certain token $Y_i \in \mathbb{R}^D$ in $Y \in \mathbb{R}^{N \times D}$ (e.g., the $i$-th token) is an integration weighted sum of all dimensions in $W^{(2)} \in \mathbb{R}^{M \times D}$:

$$Y_i = \sum_{m=1}^{M} \mathbf{z}_{i,m} W_m^{(2)}, \quad \mathbf{z} = \text{gelu}(X'W^{(1)}), \tag{4}$$

where $\mathbf{z} \in \mathbb{R}^{N \times M}$ can be referred to as integration weights within the FFN, and it can be observed that each token $Y_i$ also possesses its own integration weights $\mathbf{z}_i \in \mathbb{R}^M$. In summary, both matrices $\mathbf{a}$ and $\mathbf{z}$ represent potential locations where the assumption of information integration may hold.

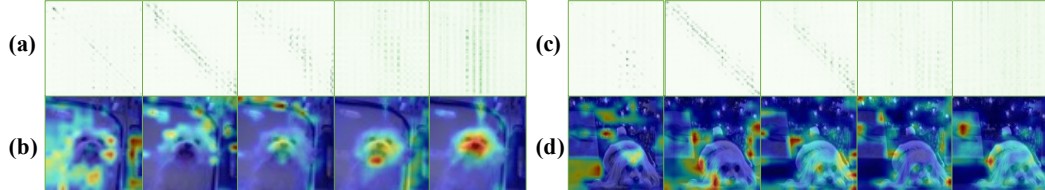

Figure 2: Visual comparison of integration weights **a** in MHSA. (a) and (b) respectively present visualizations of weights **a** at different depths of blocks for high-confidence images and the overlay of reshaped and resized rows of **a** onto the original image. Similarly, (c) and (d) depict visualizations of weights **a** for low-confidence images and their overlay onto the original image.

## 4 Information Integration Hypothesis based Transformer Diagnosis

In this section, we delve into the potential existence of information integration hypothesis mentioned in Section 3 within the MHSA and FFN. Through extensive experimental analysis, we have gathered empirical evidence supporting the hypothesis, namely dynamic integration of information among tokens and static integration of information within tokens.

### 4.1 Inter-token Information Dynamic Integration

To explore the potential existence of the information integration hypothesis within the MHSA, we analyzed the integration weights **a** in Eqn. (2). Fig. 2(a) illustrates the magnitudes of integration weights **a** in blocks of different depths within the model. It can be observed that for high-confidence samples, the integration weights **a** in shallower blocks exhibit a diagonal pattern. However, as the depth of the block increases, the integration weights **a** display a vertical pattern, consistent with observations from prior studies [63–66]. Additionally, we observed that for different high-confidence samples, the positions of the vertical lines in the integration weights **a** within the deeper blocks vary, as shown in Fig. 8 in the Appendix. This indicates that in the initial stages, MHSA primarily mixes and processes information between adjacent tokens. In the later stages, MHSA dynamically and selectively integrates specific information among tokens.

Continuing, we extracted an arbitrary row $\mathbf{a}_i$ from the integration weights **a** within the deeper blocks and removed the [CLS] token. After reshaping and resizing, we overlaid it onto the original image to generate a heatmap, as shown in Fig. 2(b). From the heatmap, it is evident that the positions of the vertical lines mainly concentrate on the foreground of the input image. This suggests that in the advanced stages of the model, MHSA primarily integrates specific information among tokens containing foreground elements. However, for low-confidence samples, as depicted in Fig. 1(c) and (d), the deeper layers of the MHSA erroneously integrate information corresponding to the background tokens. Additional quantitative and qualitative analyses of the integration weights **a** are presented in Appendices C and B. In summary, we have identified the first evidence of the existence of the information integration hypothesis in Transformer models:

***Evidence 1:*** *In the initial stages of the Transformer, MHSA primarily mixes and processes adjacent patch information among tokens. However, in the advanced stages of the Transformer, MHSA dynamically and selectively integrates specific patch information among tokens. When integrating incorrect information among tokens, termed conjunction errors, it leads to model mispredictions. We refer to this integration as inter-token information dynamic integration. The term "dynamic" arises from the fact that the $V$ to be integrated, as specified in Eqn. (2), varies with each sample.*

### 4.2 Intra-token Information Static Integration

To explore the potential existence of the information integration hypothesis within the FFN, we conducted visual analysis of the integration weights **z** in shallow and deep blocks, as depicted in Fig. 3. Fig. 3(b) illustrates the patterns of integration weights **z** in shallow and deep blocks for high-confidence samples from different classes. It can be observed that the patterns of integration weights differ between shallow and deep blocks for samples from different classes. However, in Fig. 3(a), for high-confidence samples from the same category, the patterns of integration weights **z** in shallow blocks differ, while the patterns in deep blocks show less variability and remain relatively

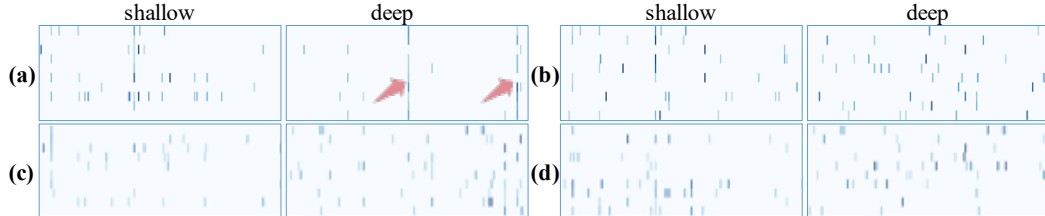

Figure 3: Visual comparison of integration weights $\mathbf{z}$ in FFN. (a) and (b) respectively illustrate visualizations of weights $\mathbf{z}$ for high-confidence samples of same classes and the different class in shallow and deep blocks. Similarly, (c) and (d) depict visualizations of weights $\mathbf{z}$ for low-confidence samples of different classes and the same class in shallow and deep blocks. Each row in the image represents a sample, and each column represents a dimension.

consistent. This indicates that in the initial stages of the model, FFN mixes and processes various category information within tokens. However, in the advanced stages, FFN statically selectively integrates specific category information within tokens.

Furthermore, as depicted in Fig. 3(d), even for samples from the same class, the patterns of integration weights $\mathbf{z}$ in deep blocks differ for low-confidence images. This suggests that when the deep FFN fails to correctly integrate specific category information within tokens, it adversely affects the model's predictions. Additional quantitative and qualitative analyses of the integration weights $\mathbf{z}$ are presented in Appendices E and D. In summary, we have identified the second evidence of the information integration hypothesis in Transformers:

*Evidence 2: In the initial stages of the Transformer, the FFN primarily mixes and processes various low-level information within tokens. However, in the advanced stages of the Transformer, the FFN statically and selectively integrates specific category information within tokens. When integrating incorrect information within tokens, termed conjunction errors, it leads to model mispredictions. The term "static" arises from the fact that the $W^{(2)}$ to be integrated, as specified in Eqn. (4), is fixed relative to each sample, and for samples of the same class, the integration weights $\mathbf{z}$ are also fixed.*

## 5 Information Integration Hypothesis based Transformer Treatments

In this section, we propose heuristic dynamic integration constraints and rule-based static integration constraints to correct conjunction errors in information integration, aiming to enhance model performance.

### 5.1 Heuristic Information Dynamic Integration Therapy

In order to alleviate conjunctive errors in dynamic integration of information among tokens, we heuristically constrain the integration weights $\mathbf{a}$ by highlighting the foreground of images with low confidence scores. Furthermore, unlike the single-head integration weights $\mathbf{a}$ in Eqn. (2), we have improved the calculation of integration weights for multi-head scenarios using gradients.

Specifically, for a Transformer classification model with $K(K \geq 2)$ classes, let $k \in \{1, 2, ..., K\}$ represent the true label for input, and $p \in \mathbb{R}^K$ denote the predicted probabilities of all classes by the Transformer. We incorporate gradients related to the true class to discern the importance of each head out of $H$ heads in self-attention, thereby obtaining the integration weights $\hat{\mathbf{a}} \in \mathbb{R}^{N \times N}$ in the multi-head scenario:

$$\hat{\mathbf{a}} = \frac{1}{H} \sum_{h=1}^{H} (\max(\frac{\partial p_k}{\partial \mathbf{a}^h}, 0) \odot \mathbf{a}^h), \tag{5}$$

where $\max(.)$ denotes setting negative values in the derivative to zero, thereby considering only the positive impact on the predicted probability of the true class. The introduction of gradients in Eqn. (5) not only helps discern which head is important but also establishes a connection between integration weights and specific classes, thereby making the weights reflecting the integration of information among tokens more accurate. Details of the comparison experiments on the introduction of gradients

can be found in Section 6.2. Next, for foreground annotation $t \in \mathbb{R}^N$ with low confidence, we constrain the integration of background information using the loss function $\mathcal{L}_{IDI}$:

$$\mathcal{L}_{IDI} = \sum_{j=1}^{N} ((\sum_{i=1}^{N} \hat{\mathbf{a}}_{i,j}) \odot (1 - t_j)), \tag{6}$$

where $t \in \mathbb{R}^N$ is a binary annotation, with 1 and 0 representing the presence and absence of foreground within the token, respectively.

## 5.2 Rule-based Information Static Integration Therapy

To correct conjunctive errors in static integration of information within tokens, we first establish integration rules within tokens when the model prediction is correct, and then constrain the integration weights $\mathbf{z}$ based on these rules. Additionally, we have also improved the integration weights $\mathbf{z}$ using gradients, establishing a connection with the true class $k$, resulting in the new integration weights $\hat{\mathbf{z}} \in \mathbb{R}^{N \times M}$ as follows:

$$\hat{\mathbf{z}} = \max(\frac{\partial p_k}{\partial \mathbf{z}}, 0) \odot \mathbf{z}. \tag{7}$$

Next, we select $S$ high-confidence samples for each class to calculate the average integration weights $\bar{\mathbf{z}} \in \mathbb{R}^{N \times M}$, and then establish binary integration rules $r \in \mathbb{R}^{N \times M}$ for each class using a threshold $\tau$:

$$r = \mathbb{1}(\bar{\mathbf{z}} \geq \tau), \bar{\mathbf{z}} = \frac{1}{S} \sum_{s=1}^{S} \hat{\mathbf{z}}_{(s)}. \tag{8}$$

The values in $r$ are 1 or 0, indicating the integration and non-integration of information in the corresponding dimension, respectively. Finally, based on the integration rules $r$ for a certain class, we enforce that erroneous information within tokens is not integrated using the loss function $\mathcal{L}_{ISI}$:

$$\mathcal{L}_{ISI} = \sum_{m=1}^{M} \sum_{n=1}^{N} (\hat{\mathbf{z}}_{n,m} \odot (1 - r_{n,m})), \tag{9}$$

During actual enforcement, each training sample needs to be constrained using the integration rules corresponding to its true class.

## 5.3 Joint Therapy of Dynamic and Static Integration

The loss functions $\mathcal{L}_{IDI}$ and $\mathcal{L}_{ISI}$ can be individually combined with the original loss function or used jointly:

$$\mathcal{L}_{total} = \mathcal{L}_{ori} + \alpha \mathcal{L}_{IDI} + \beta \mathcal{L}_{ISI}, \tag{10}$$

where $\alpha$ and $\beta$ are used to balance the magnitudes of the loss functions. In our practical experiments, we found that using the loss functions $\mathcal{L}_{IDI}$ and $\mathcal{L}_{ISI}$ sequentially yielded the most effective results. It is important to note that the therapy model only rectifies conjunctive errors in the Transformer without altering its architecture or operational procedure. Thus, during inference, it does not incur any additional computational overhead.

# 6 Experiments

## 6.1 Experimental Settings

**Datasets and Transformer Architectures.** To validate the effectiveness of Transformer Doctor, we conducted experiments on five mainstream datasets: CIFAR-10 [67], CIFAR-100 [67], ImageNet-10 [68], ImageNet-50 [69], and ImageNet-1k [68]. Furthermore, we performed experiments on various Transformer architectures used for visual classification tasks, including DeiT [48], CaiT [3], TNT [4], PVT [5], Eva [6], and BeiT [49], in addition to ViT [2]. It is important to note that Transformer Doctor diagnoses and treats already trained Transformer models. More experimental settings and results can be found in Appendix.

**Parameter Settings.** During all training stage, each dataset was trained for 300 epochs using the AdamW [70] optimizer, with an initial learning rate of 0.01. The learning rate decayed according to a

Table 1: Performance of Transformer Doctor on various SOTA Transformers. '+Doctor' indicates the performance of model treated with Transformer Doctor (All Score are in %).

| | CIFAR-10 | CIFAR-100 | ImageNet-10 | ImageNet-50 | ImageNet-1K |
|---|---|---|---|---|---|
| **ViT-Tiny** | 82.17 | 56.02 | 78.8 | 59.62 | 64.77 |
| *+Doctor* | 83.00 (+0.87) | 58.08 (+2.06) | 80.80 (+2.00) | 61.02 (+1.40) | 68.86 (+4.09) |
| **DeiT-Tiny** | 82.71 | 56.97 | 80.20 | 61.47 | 66.83 |
| *+Doctor* | 83.96 (+1.25) | 59.49 (+2.52) | 81.20 (+1.00) | 63.19 (+1.69) | 70.75 (+3.92) |
| **CaiT-XXS** | 82.64 | 56.10 | 75.80 | 58.32 | 66.28 |
| *+Doctor* | 84.20 (+1.36) | 60.00 (+3.90) | 77.80 (+2.00) | 60.16 (+1.84) | 70.25 (+3.97) |
| **TNT-Small** | 83.31 | 54.67 | 81.60 | 65.45 | 67.25 |
| *+Doctor* | 84.33 (+1.02) | 55.60 (+0.93) | 83.00 (+1.40) | 67.64 (+2.08) | 69.97 (+2.72) |
| **PVT-Tiny** | 83.27 | 51.94 | 82.00 | 71.81 | 67.73 |
| *+Doctor* | 84.82 (+1.55) | 55.10 (+3.16) | 84.20 (+2.20) | 74.53 (+2.72) | 70.94 (+3.21) |
| **Eva-Tiny** | 87.56 | 64.15 | 83.80 | 71.23 | 72.51 |
| *+Doctor* | 88.28 (+0.72) | 64.99 (+0.84) | 85.80 (+2.00) | 72.95 (+1.72) | 75.45 (+2.94) |
| **BeiT-Tiny** | 74.69 | 49.58 | 79.80 | 71.59 | 70.46 |
| *+Doctor* | 76.20 (+1.51) | 51.03 (+1.45) | 82.20 (+2.40) | 73.57 (+1.98) | 71.98 (+3.12) |

cosine annealing schedule, with T_max set to 300 epochs. Additionally, $\alpha$ and $\beta$ were set to default values of 10 and 100, respectively, to balance each loss function. The default value of $\tau$ was 0.15, and the constrained loss function was applied by default to the last block.

**Baseline Models.** In addition to using the original pre-trained Transformer models as baselines, as shown in Table 1, we also established a blank control group, which involves no method introduction but continues training for the same epochs, as presented in Table 3. Furthermore, we compared different ways of integrating multi-head integration weights in Eqn. 5 and not introducing gradients in Eqn. 7 as simple baselines against the proposed final method, as illustrated in Table 2. Due to differences in computational resources, certain training configurations, such as batch size, differ from those in the original work, leading to slight variations in the baseline. However, all comparative experiments were conducted under fair conditions. For detailed experimental results, please refer to Table 3, and for further setup information, see Appendix F.

## 6.2 Quantitative Analysis

**The Performance of Transformer Doctor on SOTA Models.** We evaluated the performance of Transformer Doctor across five major datasets and seven mainstream architectures. As shown in Table 1, it is evident that Transformer Doctor effectively enhances model performance on both small and large-scale datasets. Specifically, after treatment, the accuracy of CaiT improved by 1.36% on CIFAR-10 and 1.84% on ImageNet-50. The BeiT model, after treatment, saw an accuracy increase of 1.45% on CIFAR-100 and 2.39% on ImageNet-50. Additionally, we observed that generally, the larger the dataset, the more significant the performance improvement by Transformer Doctor. For instance, the accuracy improvement of ViT on ImageNet-1K was 2.09% higher than on ImageNet-10. The reason for this could be that, similar to biological vision, Transformer misrecognition can be due to both conjunction errors and feature extraction failures. Large-scale datasets effectively help the model learn a vast array of features, thus better extracting the input image's features. On this basis, treating conjunction errors can maximize model performance. However, for small-scale datasets, the dominance of not extracting effective features may outweigh conjunction errors. Therefore, focusing solely on treating conjunction errors is less effective on small-scale datasets compared to large-scale ones. More detailed experimental results can be found in Table 3 in the appendix.

**The Performance of Transformer Doctor with Various Computational Forms.** We compared the performance of Transformer Doctor under different computational forms represented by Eqn. (5) and Eqn. (7), as shown in Table 2. It can be observed that Transformer Doctor performs better under the computational forms of Eqn. (5) and Eqn. (7). Specifically, under dynamic integration constraints,

Table 2: Comparison of accuracy of Transformer Doctor under different computational formulations. 'mean' denotes directly averaging integration weights across all heads, 'min' and 'max' respectively represent taking the minimum and maximum integration weights within each head. Each row corresponds to ViT-Tiny and PVT-Tiny architectures, with ImageNet-10 dataset used for evaluation.

| Base | +IDI | | | | +ISI | |
|---|---|---|---|---|---|---|
| | $\min(\mathbf{a}^h)$ | $\max(\mathbf{a}^h)$ | $\mathrm{mean}(\mathbf{a}^h)$ | $\hat{\mathbf{a}}$ | $\mathbf{z}$ | $\hat{\mathbf{z}}$ |
| 78.80 | 79.20 (+0.40) | 78.60 (-0.20) | 79.20 (+0.40) | **80.20 (+1.40)** | 79.00 (+0.20) | **80.40 (+1.60)** |
| 82.00 | 81.20 (-0.80) | 81.20 (-0.80) | 80.00 (-2.00) | **83.60 (+1.36)** | 82.40 (+0.40) | **83.40 (+1.40)** |

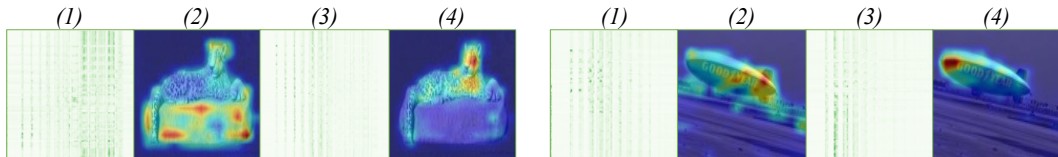

Figure 4: Comparison of inter-token integration weights before and after introducing Transformer Doctor. (1) and (3) depict integration weights before and after treatment, respectively. (2) and (4) show the corresponding heat map effects of integration weights overlaid onto the original image before and after treatment.

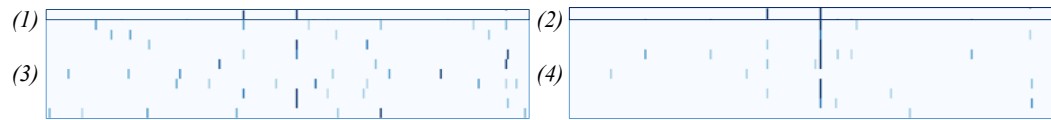

Figure 5: Comparison of intra-token integration weights before and after introducing Transformer Doctor. (1) and (2) represent the intra-token integration rules for correct predictions. (3) and (4) depict the intra-token integration weights before and after treatment, respectively.

taking the minimum, maximum, or averaging fusion of integration weights for each head does not significantly improve model performance. For instance, when taking the maximum value, the accuracy of PVT-Tiny decreases by 0.08% compared to the base. However, under the computational form of Eqn. (5), the accuracy of PVT-Tiny improves by 2%. Under static integration constraints, the accuracy of the model under the original integration weight calculation form is 79.00%, while the accuracy under the computational form of Eqn. (7) is 80.40%.

### 6.3 Qualitative Analysis

**Comparison of the Intra-token Integration Weights a.**

We also compared the static integration weights within tokens before and after treatment, as shown in Fig. 5. From panels (1) and (3), it is apparent that before treatment, the static integration weights for misclassified samples do not adhere to the correct prediction rules. However, panels (2) and (4) in Fig. 5 demonstrate that after treatment, the static integration weights align with the correct prediction rules, integrating the correct class information, which results in accurate predictions.

**Comparison of the Inter-token Integration Weights z.** In Fig. 5, we compared the static integration weights within tokens before and after treatment. As shown in Fig. 5 (1), before treatment, the static integration weights of misclassified samples do not adhere to the correct integration rules for prediction, i.e., integration of incorrect category information within tokens. However, as depicted in Fig. 5 (2), after treatment, the static integration weights conform to the correct integration rules for prediction, integrating the correct category information within tokens.

# 7   Conclusion

This paper introduces the first framework, Transformer Doctor, designed for diagnosing and treating internal errors within Transformers simultaneously. Specifically, distinct from existing post-hoc interpretability methods for Transformers, this work draws inspiration from information integration and conjunctive errors in the biological visual system, proposing and validating the hypothesis of information integration for Transformers. Furthermore, addressing conjunctive errors within information integration, this paper presents corresponding error treatment methods. Extensive qualitative and quantitative analyses conducted on mainstream datasets and Transformer architectures demonstrate the effectiveness and applicability of Transformer Doctor.

**Limitations and Future Work.** It is undeniable that we have only validated Transformer Doctor on mainstream visual recognition tasks, leaving more complex machine vision tasks for further exploration. Furthermore, investigating whether the information integration hypothesis holds true in the field of natural language processing or multimodal domains is also a worthwhile research endeavor. Additionally, exploring more error mechanisms and developing a more automated and intelligent Transformer Doctor framework are the focal points of our future work.

## Acknowledgments and Disclosure of Funding

This work is supported by National Natural Science Foundation of China (62376248), Ningbo Natural Science Foundation (2022J182) and the Fundamental Research Funds for the Central Universities (226-2024-00145).

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

# Appendix

In the appendix, we offer additional evidence concerning the information integration mechanism and conjunction errors in the Transformer. Furthermore, more quantitative and qualitative analysis experiments on Transformer Doctor are provided. Additionally, the algorithm code for the Transformer Doctor is included in the uploaded *source_codes.zip* file.

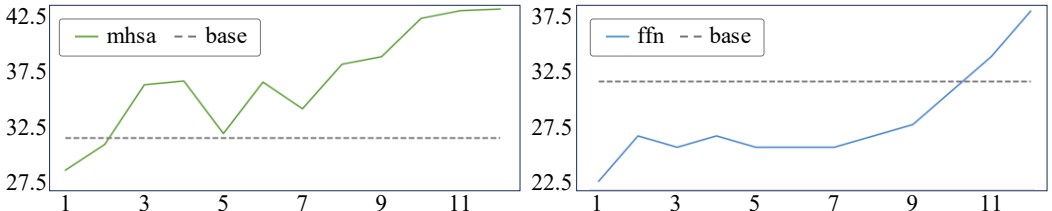

Figure 6: Comparison of model accuracy on low-confidence samples from ImageNet-10 after replacing original integration weights with ideal integration weights in ViT. (a) and (b) depict the accuracy line plots of the model with ideal integration weights replacing original integration weights step by step from shallow to deep blocks in MHSA and FFN, respectively. The ideal integration weights in MHSA are manually annotated for foreground regions in images, while those in FFN are statistically generated by integrating weights of high-confidence images for each class.

## A    Broader Impact

This paper contributes to the research field of understanding and optimizing Transformers. Specifically, drawing inspiration from biological vision, the exploration of error mechanisms in Transformers for machine vision tasks provides a novel perspective for understanding the internal workings of Transformers. Additionally, the interpretable treatment solutions proposed based on errors in Transformers offer a new pathway for optimizing Transformer models.

## B    Additional Visual Evidence of Inter-token Information Dynamic Integration

As illustrated in Fig. 7, we provide further visual comparisons of integration weights **a** with high and low confidences to demonstrate the dynamic information integration mechanism among tokens in the MHSA. The observed phenomena align with our conclusions in the main text. Specifically, in the early stages of the Transformer, as shown in Fig. 7(a), the MHSA predominantly blends and processes information among adjacent tokens, whereas in the advanced stages, the MHSA dynamically and selectively integrates specific information among tokens. Simultaneously, for low-confidence samples, as depicted in Fig. 7(b), the advanced-stage MHSA erroneously integrates token information corresponding to the background.

In addition, we provide more visualizations of integration weights **a** for high-confidence images in the final block, as shown in Fig. 8. It can be observed that the integration weights **a** vary with different input samples. This indicates that in the advanced stage, the integration weights **a** dynamically integrate specific information for the final prediction.

## C    Quantitative Analysis of Conjunction Errors in Inter-token Information Dynamic Integration

To verify whether incorrect predictions are indeed caused by the Transformer integrating erroneous information among tokens in the advanced stage, we selected low-confidence samples from the test set and evaluated the model's performance after replacing the original integration weights with ideal ones, as shown in Fig. 6(a). It can be observed that forcibly integrating the correct information among

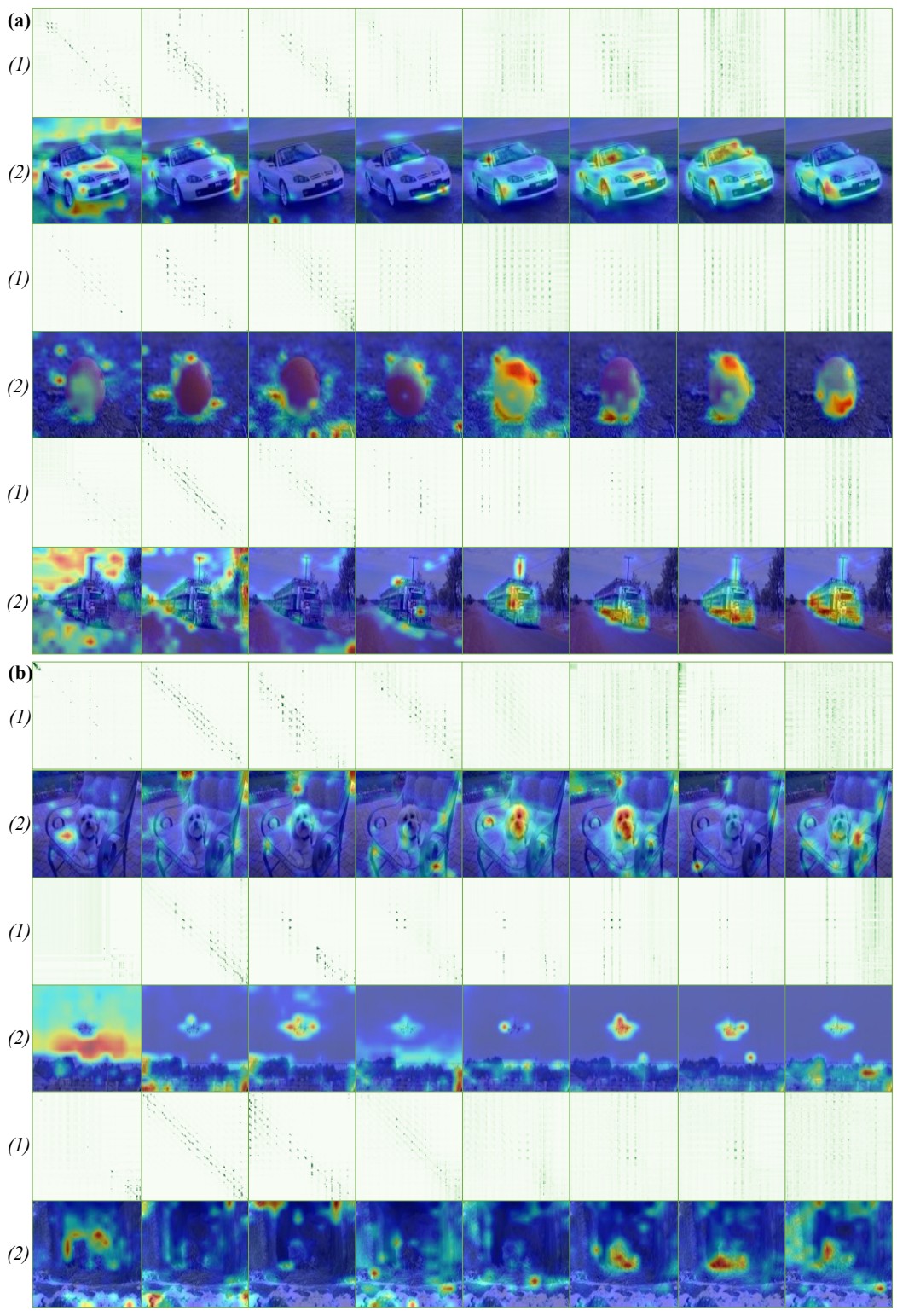

Figure 7: Visualization comparison of integration weights **a** in MHSA. (a) and (b) correspond to high-confidence and low-confidence samples, respectively. (1) and (2) show the visualizations of integration weights **a** in blocks from shallow to deep from left to right, as well as the visualization of reshaped and resized rows of **a** superimposed onto the original image.

|     (1)     |     (2)     |     (1)     |     (2)     |     (1)     |     (2)     |     (1)     |     (2)     |

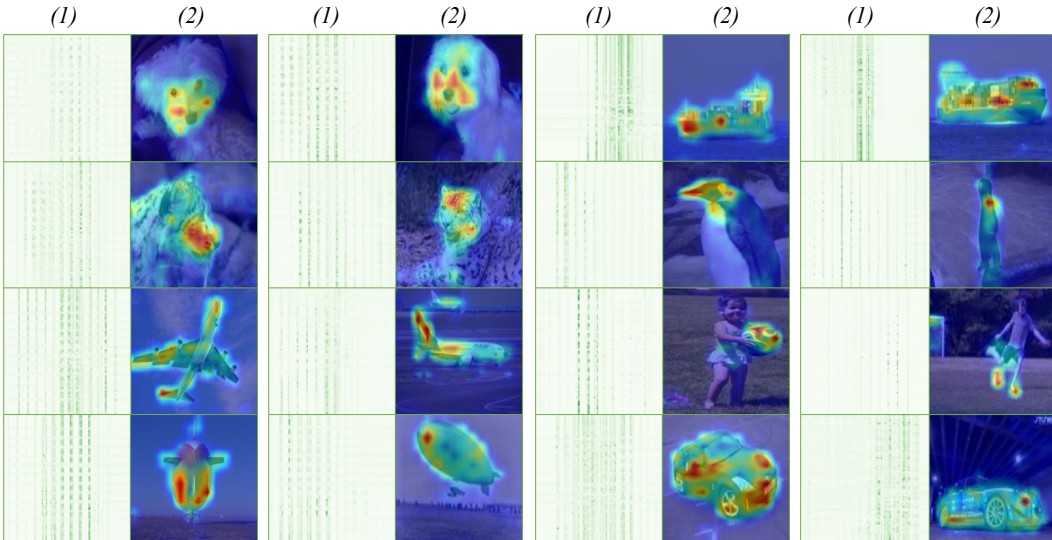

Figure 8: Visualization comparison of integration weights **a** in the last block of MHSA. (1) and (2) depict the visualization of **a** and the visualization of reshaped and resized rows of **a** superimposed onto the original image.

tokens in the model's early stages does not improve performance. This is because the model in its early stages is merely blending and processing various information, rather than integrating specific information. However, when correct information is forcibly integrated among tokens in the advanced stages, the model's performance improves significantly.

## D    Additional Visual Evidence of Intra-token Information Static Integration

As illustrated in Fig. 9, we provide further visual comparisons of integration weights **z** with high and low confidences to demonstrate the static information integration mechanism within tokens in the FFN. The observed phenomena align with our conclusions in the main text. Specifically, in the early stages of the Transformer, as shown in Fig. 9(a), the FFN mixes and processes various categories of information within tokens. However, in the advanced stages, the FFN statically and selectively integrates specific category information within tokens. Additionally, for low-confidence samples, as depicted in Fig. 9(b), the FFN in the advanced stages integrates incorrect category information within tokens.

## E    Quantitative Analysis of Conjunction Errors in Intra-token Information Static Integration

To verify whether incorrect predictions are indeed caused by the FFN integrating erroneous information within tokens in the advanced stage of the Transformer, we selected low-confidence samples from the test set and evaluated the model's performance after replacing the original integration weights with ideal ones, as shown in Fig. 6(b). The conclusions are similar to those observed in the dynamic information integration among tokens: forcibly integrating correct information within tokens in the model's early stages does not improve performance. This is because, in the early stages, the model is merely mixing and processing various types of information rather than integrating specific information. However, when correct information is forcibly integrated within tokens in the advanced stages, the model's performance improves significantly.

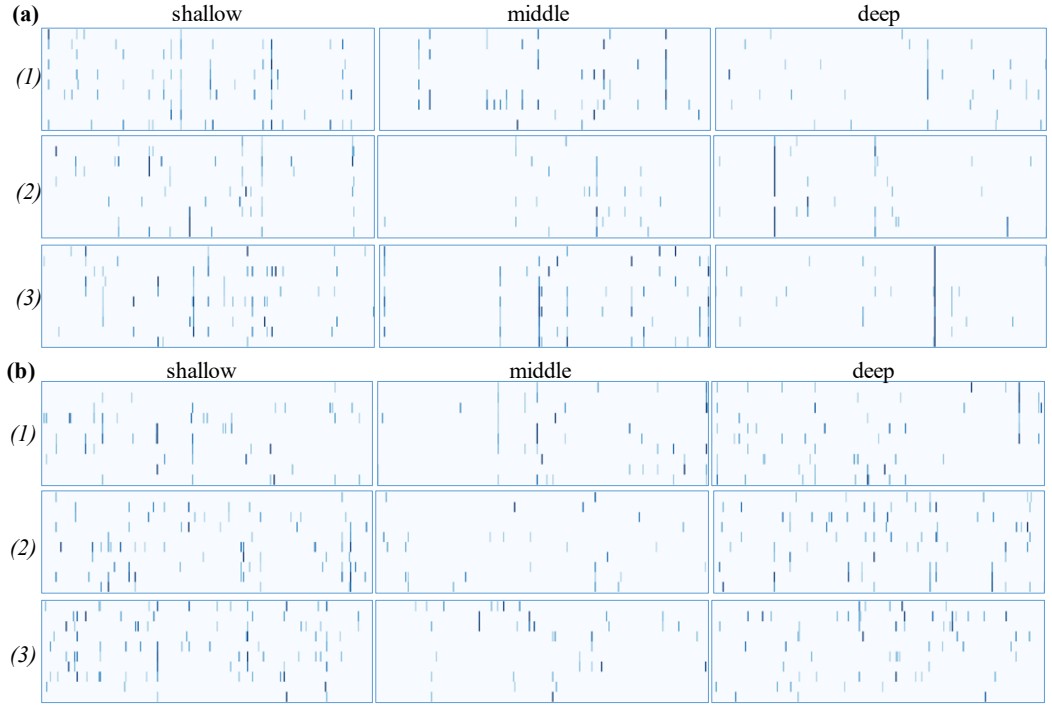

Figure 9: Visualization comparison of integration weights **z** in the FFN. (a) and (b) correspond to high-confidence and low-confidence samples, respectively. (1), (2), and (3) represent the visualization of integration weights **z** in blocks from shallow to deep for three classes. Each row of each image represents different samples of the class, and each column represents different dimensions of the integration weights **z**.

## F  Additional Experimental Settings

To validate the dynamic integration constraints mentioned in Section 5.1, we manually annotated the foreground masks for the 10 lowest-confidence samples in each class of ImageNet-10. For ImageNet-50 and ImageNet-1000, we utilized the segmentation annotations provided in the ImageNet-S dataset. This dataset includes segmentation masks for 10 randomly selected samples per class in ImageNet-50 and for 10 randomly selected samples per class in 919 classes of ImageNet-1000. For CIFAR-10 and CIFAR-100, due to the small size of the images making it difficult to annotate the foreground, we only used these datasets to validate the static integration constraints mentioned in Section 5.2. Additionally, within the static integration constraints, considering that some Transformers utilize the [CLS] token for visual recognition tasks, we constrained only the static integration within the [CLS] token. In the experiments, we utilized two Linux servers, each equipped with 8 NVIDIA A6000 GPU cards, 24 CPU cores, and 500GB of memory.

## G  Additional Results of Transformer Doctor on SOTA Transformers

Table 3 presents additional experimental results of Transformer Doctor on various mainstream datasets and architectures, including blank control groups, solely introducing dynamic integration constraints, solely introducing static integration constraints, and jointly introducing dynamic and static integration constraints. It is evident that both solely and jointly introducing dynamic or static integration constraints significantly enhance model performance. Specifically, when solely introducing dynamic integration constraints, the accuracy of CaiT-XXS increased by 1.2% on ImageNet-10 and 1.57% on ImageNet-1K. When solely introducing static integration constraints, TNT-Tiny saw accuracy improvements of 1.02% on CIFAR-10 and 1.58% on ImageNet-50. When jointly introducing dynamic and static integration constraints, the ViT-Tiny model's accuracy increased by 1.40% on ImageNet-50 and 4.09% on ImageNet-1K.

Table 3: Comparison of accuracy of Transformer Doctor across various SOTA models. '+Blank' refers to a blank control model trained for the same number of epochs without introducing any constraints. '+IDI', '+ISI', and '+IDI, ISI' denote models with individually introduced dynamic integration constraint, individually introduced static integration constraint, and simultaneously introduced dynamic and static integration constraints, respectively. Due to the small size of CIFAR-10 and CIFAR-100 images, which makes foreground annotation challenging, experiments involving dynamic integration constraints were not conducted. Additionally, we replaced the results of experiments simultaneously introducing dynamic integration and static integration constraints with those of experiments solely involving static integration constraints. (all scores are in %)

| | CIFAR-10 | CIFAR-100 | ImageNet-10 | ImageNet-50 | ImageNet-1K |
|---|---|---|---|---|---|
| **ViT-Tiny** | 82.17 | 56.02 | 78.8 | 59.62 | 64.77 |
| +Blank | 82.22 (+0.05) | 56.05 (+0.03) | 78.90 (+0.10) | 59.71 (+0.09) | 64.74 (-0.03) |
| *+IDI* | - | - | 80.20 (+1.40) | 60.45 (+0.83) | 66.42 (+1.65) |
| *+ISI* | 83.00 (+0.87) | 58.08 (+2.06) | 80.40 (+1.60) | 60.89 (+1.27) | 66.33 (+1.56) |
| *+IDI, ISI* | 83.00 (+0.87) | 58.08 (+2.06) | 80.80 (+2.00) | 61.02 (+1.40) | 68.86 (+4.09) |
| **DeiT-Tiny** | 82.71 | 56.97 | 80.20 | 61.47 | 66.83 |
| +Blank | 82.69 (-0.02) | 56.89 (-0.08) | 80.20 (+0.00) | 61.57 (+0.10) | 66.86 (+0.03) |
| *+IDI* | - | - | 81.00 (+0.80) | 62.60 (+1.13) | 68.20 (+1.37) |
| *+ISI* | 83.96 (+1.25) | 59.49 (+2.52) | 80.60 (+0.40) | 62.16 (+0.69) | 68.10 (+1.24) |
| *+IDI, ISI* | 83.96 (+1.25) | 59.49 (+2.52) | 81.20 (+1.00) | 63.19 (+1.69) | 70.75 (+3.92) |
| **CaiT-XXS** | 82.64 | 56.10 | 75.80 | 58.32 | 66.28 |
| +Blank | 82.64 (+0.00) | 56.08 (-0.02) | 75.70 (-0.10) | 58.33 (+0.01) | 66.38 (+0.10) |
| *+IDI* | - | - | 77.20 (+1.40) | 58.95 (+0.63) | 67.43 (+1.15) |
| *+ISI* | 84.20 (+1.36) | 60.00 (+3.90) | 77.00 (+1.20) | 59.90 (+1.58) | 67.85 (+1.57) |
| *+IDI, ISI* | 84.20 (+1.36) | 60.00 (+3.90) | 77.80 (+2.00) | 60.16 (+1.84) | 70.25 (+3.97) |
| **TNT-Small** | 83.31 | 54.67 | 81.60 | 65.45 | 67.25 |
| +Blank | 83.27 (-0.04) | 54.74 (+0.07) | 81.60 (+0.00) | 65.45 (+0.00) | 67.23 (-0.02) |
| *+IDI* | - | - | 82.60 (+1.00) | 67.33 (+1.88) | 68.34 (+1.09) |
| *+ISI* | 84.33 (+1.02) | 55.60 (+0.93) | 82.20 (+0.60) | 67.03 (+1.58) | 68.52 (+1.27) |
| *+IDI, ISI* | 84.33 (+1.02) | 55.60 (+0.93) | 83.00 (+1.40) | 67.64 (+2.08) | 69.97 (+2.72) |
| **PVT-Tiny** | 83.27 | 51.94 | 82.00 | 71.81 | 67.73 |
| +Blank | 83.32 (+0.05) | 51.99 (+0.05) | 81.20 (+0.20) | 71.79 (-0.02) | 67.74 (+0.01) |
| *+IDI* | - | - | 83.60 (+1.60) | 74.06 (+2.25) | 69.15 (+1.42) |
| *+ISI* | 84.82 (+1.55) | 55.10 (+3.16) | 83.40 (+1.40) | 74.18 (+2.37) | 68.99 (+1.26) |
| *+IDI, ISI* | 84.82 (+1.55) | 55.10 (+3.16) | 84.20 (+2.20) | 74.53 (+2.72) | 70.94 (+3.21) |
| **Eva-Tiny** | 87.56 | 64.15 | 83.80 | 71.23 | 72.51 |
| +Blank | 87.62 (+0.06) | 64.10 (-0.05) | 83.82 (+0.00) | 71.24 (+0.02) | 72.58 (+0.07) |
| *+IDI* | - | - | 85.20 (+1.40) | 72.24 (+1.01) | 74.08 (+1.57) |
| *+ISI* | 88.28 (+0.72) | 64.99 (+0.84) | 85.20 (+1.40) | 72.57 (+1.34) | 73.85 (+1.34) |
| *+IDI, ISI* | 88.28 (+0.72) | 64.99 (+0.84) | 85.80 (+2.00) | 72.95 (+1.72) | 75.45 (+2.94) |
| **BeiT-Tiny** | 74.69 | 49.58 | 79.80 | 71.59 | 70.46 |
| +Blank | 74.66 (-0.03) | 49.63 (+0.05) | 79.79 (-0.01) | 71.58 (-0.01) | 70.51 (+0.05) |
| *+IDI* | - | - | 81.40 (+1.60) | 72.72 (+1.13) | 72.21 (+1.75) |
| *+ISI* | 76.20 (+1.51) | 51.03 (+1.45) | 80.60 (+0.80) | 72.98 (+1.39) | 71.72 (+1.26) |
| *+IDI, ISI* | 76.20 (+1.51) | 51.03 (+1.45) | 82.20 (+2.40) | 73.57 (+1.98) | 71.98 (+3.12) |

# H   Additional Visual Comparisons of Integration Weight a Before and After Treating Conjunction Errors

As shown in Fig. 10, we compare additional dynamic integration weights **a** before and after treatment. The results align with our observations in the main text. Specifically, from Fig. 4(1, 2), it can be seen that before treatment, the dynamic integration weights of misclassified samples exhibit irregular

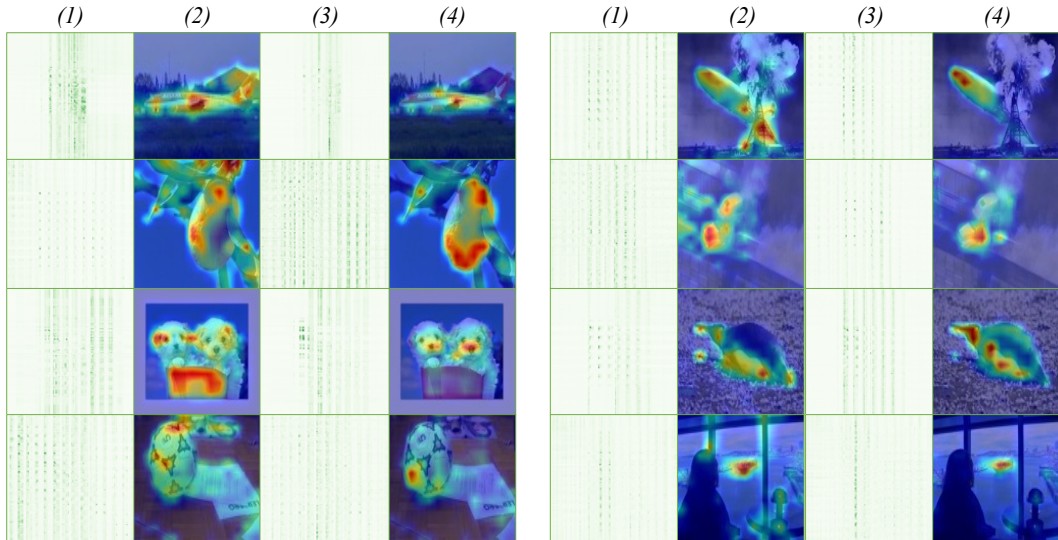

Figure 10: Comparison of dynamic integration weights among tokens before and after introducing Transformer Doctor. (1) and (3) show the integration weights before and after treatment, respectively. (2) and (4) illustrate the corresponding heatmap effects of these integration weights superimposed onto the original image, where higher brightness indicates larger weight values. The experiments were conducted on ViT and ImageNet-10.

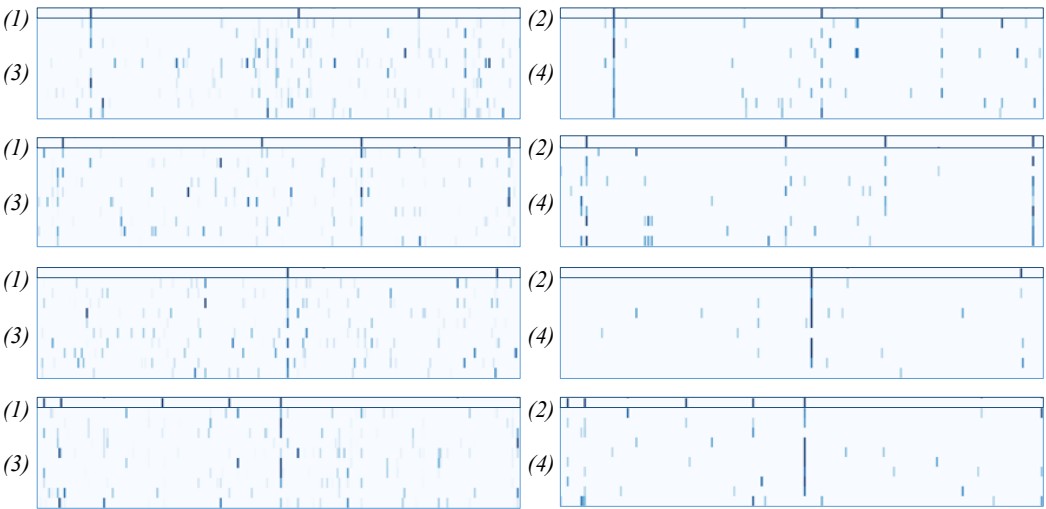

Figure 11: Comparison of static integration weights within tokens before and after Transformer Doctor diagnosis and treatment. (1) and (2) illustrate the static integration rules during correct predictions. (3) and (4) show the static integration weights before and after treatment, respectively. The experiments were conducted on ViT and ImageNet-10.

distributions, indicating the integration of erroneous background information among tokens. However, after treatment, as illustrated in Fig. 4(3, 4), the integration weights selectively integrate the correct foreground information.

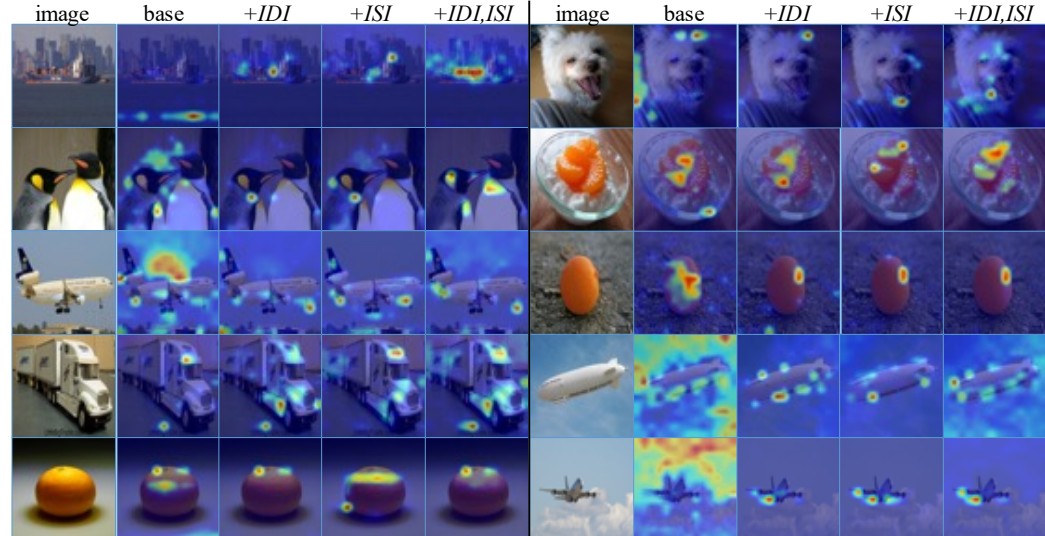

Figure 12: Comparison of feature attribution maps before and after introducing Transformer Doctor. '+IDI', '+ISI', and '+IDI, ISI' refer to models with individually introduced dynamic integration constraint, individually introduced static integration constraint, and jointly introduced dynamic and static integration constraints, respectively.

Table 4: Comparison of model accuracy with different threshold values $\tau$ for static integration constraints within tokens. Each row corresponds to ViT-Tiny and PVT-Tiny architectures, using the ImageNet-10 dataset.

| Base | +ISI | | | | | | | | | | |
|------|------|------|------|------|------|------|------|------|------|------|------|
| | 0.0 | 0.1 | 0.2 | 0.3 | 0.4 | 0.5 | 0.6 | 0.7 | 0.8 | 0.9 | 1.0 |
| 78.8 | 78.8 | 80.0 | 80.4 | 79.0 | 77.8 | 78.2 | 78.0 | 78.6 | 78.4 | 78.0 | 78.0 |
| 82.0 | 82.0 | 82.2 | 83.6 | 83.2 | 82.4 | 82.6 | 81.8 | 81.8 | 80.8 | 81.0 | 81.2 |

# I  Additional Visual Comparisons of Integration Weight z Before and After Treating Conjunction Errors

As depicted in Fig. 11, we present additional comparisons of integration weights **z** before and after treatment. The results are consistent with our observations in the main text. Specifically, from Fig. 11(1, 2), it can be observed that before treatment, the static integration weights of misclassified samples do not adhere to the correct integration rules, indicating the integration of incorrect category information within tokens. However, in Fig. 11(3, 4), it can be seen that after treatment, the static integration weights adhere to the correct integration rules, integrating fewer incorrect information, thus improving the model performance.

# J  Comparison of Feature Attribution Maps via Rollout

We also visualized the attribution maps of model predictions using Rollout feature attribution techniques, as shown in Fig. 12. From the figure, it can be observed that without introducing any constraints, the model tends to focus on unnecessary background information. However, when dynamic integration constraints or static integration constraints are introduced, the model starts to focus on foreground information relevant to the categories. When both dynamic and static integration constraints are simultaneously introduced, the key information relied upon by the model for prediction aligns more closely with human understanding, i.e., it focuses more on the object itself.

# K  Ablation Study on Threshold $\tau$ in Eqn. (8)

We conducted ablation experiments on the threshold $\tau$ that determines the intra-token integration rule in equation (8), as shown in Table 4. Using the ViT model as an example, we observed that when the threshold $\tau$ increases from 0.0 to 0.2, the model accuracy gradually increases from 78.8% to 80.4%, exceeding the baseline. This improvement occurs because, with a very low threshold, such as 0.0, the integration rule includes redundant intra-token information weights, meaning that unnecessary intra-token information is integrated, which does not significantly enhance model performance.

Conversely, when the threshold $\tau$ increases from 0.3 to 1.0, the model accuracy gradually decreases from 80.4% back to 78.0%, falling below the baseline (note that the table presents the highest accuracy on the test set, not the final accuracy after model training). This decline is due to the overly high threshold, such as 0.8, causing the integration rule to miss certain necessary intra-token information weights, meaning that essential intra-token information is not integrated, resulting in reduced model performance.

# L  Ablation Study on Block Depth

Fig. 13 illustrates the effects of introducing Transformer Doctor at different block depths. The figure demonstrates that the model's performance generally improves with increasing block depth when applying inter-token dynamic integration constraints. When these constraints are applied before the 7th block, the model's performance does not show significant improvement and can even fall below the baseline. However, after the 7th block, the model's performance noticeably exceeds the baseline. This observation reaffirms that in the early stages, the Transformer mixes and processes adjacent token information, while in the advanced stages, it dynamically and selectively integrates specific information. Thus, inter-token dynamic integration constraints are most effective at deeper blocks and can help enhance model performance.

Similarly, the model accuracy surpasses the baseline only when the intra-token static integration constraints are applied after the 11th block. This is because the Transformer selectively and statically integrates specific intra-token information only at the advanced stages. Furthermore,

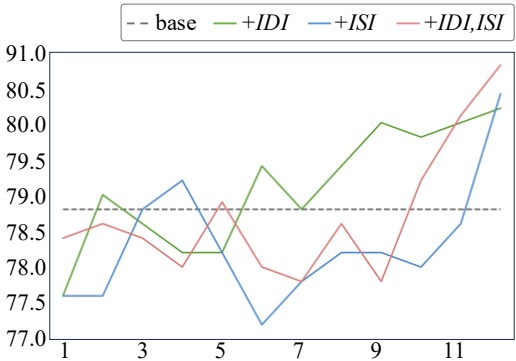

Figure 13: Comparison of accuracy when introducing Transformer Doctor in different blocks of the Transformer. Similarly, '+IDI', '+ISI', and '+IDI, ISI' refer to models with individually introduced dynamic integration constraint, individually introduced static integration constraint, and jointly introduced dynamic and static integration constraints, respectively.

the introduction of combined dynamic and static constraints results in higher model accuracy at deeper blocks compared to both the baseline and the introduction of individual integration constraints.

