# OpenReview forum: "Transformer Doctor: Diagnosing and Treating Vision Transformers"
_NeurIPS.cc/2024/Conference — NeurIPS 2024 poster_

### Official Review · Reviewer_KQuQ · 2024-07-03

**Soundness:** 4
**Presentation:** 3
**Contribution:** 4
**Rating:** 9
**Confidence:** 5

**Summary:**

This paper proposes a vision transformer diagnosing and treating framework, namely Transformer Doctor, to reveal the problems that bring about negative impact on the network performance and fix them. The paper firstly proposes the information integrating hypothesis, which argues that transformers do information refining and integrating at the lower and higher layers respectively. Based on this hypothesis, the inter-token information dynamic integration and intra-token information static integration mechaisms are designed to show the inner mechanisms of ViTs and help to treat them.

**Strengths:**

1. This paper designs a possible way to explain why a transformer cannot work well and try to fix the potential problems to improve network performance.
2. The information integration hypothesis is proposed, and two situations (self-attention and fully-connected layers) are analyzed.
3. Based on the information integration hypothesis, a transformer diagnosis and treatment method is proposed.
4. The experimental results on several databases can support the proposed method.

**Weaknesses:**

1. The evidence of the correctness of the information integration hypothesis is not strong enough, it is better to consider to prove the hypothesis in mathematic way

**Questions:**

N.A.

**Limitations:**

The authors consider the limitations of the method in the part of Conclusion.

---

> ### Author Rebuttal · Authors · 2024-08-06
>
> Thank you for your positive feedback on this work. We are pleased that you highlighted the core of our research, which is designing a potential method to explain why Transformers may not perform well and attempting to address these issues to improve network performance.
>
> We acknowledge that there are still some imperfections in our manuscript, but we have actively addressed these issues with the aim of improving the work. Below are our responses to your main comments (each of your comments is highlighted in italics).
>
> > Q1: *The evidence of the correctness of the information integration hypothesis is not strong enough, it is better to consider to prove the hypothesis in mathematic way*
> >
>
> Thank you for your review and valuable feedback. We understand your concern about the mathematical proof of the information integration hypothesis. However, our study is primarily empirical, aiming to validate the hypothesis through experimental evidence rather than mathematical derivation.
>
> Specifically, the motivation for this work stems from error mechanisms observed in biological vision systems. We chose an empirical approach to explore the practical effects and applications of the information integration hypothesis, which is a commonly used and accepted method in related research fields. In validating of the information integration hypothesis, we conducted extensive experiments and data analyses, including numerous qualitative analyses (as detailed in Sections 4.1, 4.2, Appendix B, and Appendix D) and thorough quantitative analyses (as shown in Appendix A, Appendix C, etc.). Additionally, in the error treatment based on the information integration hypothesis, we performed extensive qualitative analyses (in Sections 6.3, Appendix H, Appendix I, Appendix J) and substantial quantitative analyses (in Sections 6.2, Appendix G, Appendix K, Appendix L). These experiments provide substantial evidence for the effectiveness of the information integration hypothesis.
>
> Nevertheless, we highly value your suggestion and will emphasize in the discussion section of the paper why we chose an empirical approach and explain its advantages and limitations relative to mathematical proof. Additionally, we will attempt to supplement the theoretical background and relevant mathematical models to further support our hypothesis. We hope this response addresses your concerns.

---

> > ### Comment · Reviewer_KQuQ · 2024-08-13
> >
> > Thank you for your response, I would like to keep my original score unchange.

---

> > > ### Author Response · Authors · 2024-08-13
> > >
> > > Thank you for recognizing the value of our work. We also greatly appreciate your thorough review and insightful comments.

---

### Official Review · Reviewer_3nqh · 2024-07-08

**Soundness:** 4
**Presentation:** 4
**Contribution:** 4
**Rating:** 7
**Confidence:** 5

**Summary:**

Inspired by the information integration mechanisms and conjunction errors in the biological visual system, this paper investigates the error mechanisms within Transformers. Through a comprehensive analysis and experimental validation of the computational processes of the two core modules of Transformers, MHSA and FFN, the authors introduce the Transformer Doctor framework. The principal components of this framework are as follows:
1. Diagnosis: The paper identifies dynamic information integration among tokens within MHSA and static information integration within tokens in FFN, as well as conjunction errors occurring during the integration process.
2. Treatment: To mitigate the conjunction errors identified in MHSA and FFN, the paper proposes heuristic dynamic information integration constraints for MHSA and rule-based static information integration constraints for FFN.

To validate the efficacy of Transformer Doctor, the authors performed extensive quantitative and qualitative analyses on various Vision Transformer architectures. The findings indicate that Transformer Doctor effectively reduces conjunction errors in Transformers and enhances overall performance.

**Strengths:**

1. The investigation into the error mechanisms of Vision Transformers presented in this paper is uniquely motivated. The authors provide a novel perspective by drawing an insightful connection between error mechanisms in biological vision and machine vision. The paper conducts in-depth analyses of both MHSA and FFN, uncovering intriguing phenomena related to dynamic and static information integration, respectively.

2. The paper offers a comprehensive and detailed review of existing interpretability methods. The proposed approaches for diagnosing and treating Transformer errors are rigorous and persuasive. Extensive experiments, featuring abundant quantitative results and qualitative visual analyses, convincingly demonstrate that the proposed methods effectively mitigate errors and enhance model performance.

3. The structure of the paper is well-organized, clearly articulated, and highly accessible. Each section is meticulously arranged, providing a systematic and comprehensive flow from background and motivation to methods and experimental results.

4. This paper's comprehensive investigation into the error mechanisms of Vision Transformers addresses a relatively underexplored area in the existing literature. Drawing inspiration from error mechanisms in biological visual systems, the proposed methodologies for diagnosing and rectifying Transformer errors provide substantial insights. The elucidated integration mechanisms and identified error phenomena within Transformers are not only of considerable interest but also highly enlightening for the field.

**Weaknesses:**

1. The visualization from vertical lines to diagonals in the attention matrices presented in Fig. 2(a,c) lacks sufficient clarity. This visualization is essential for validating the authors' information integration hypothesis in MHSA. It is recommended that the authors refer to similar visualization results in studies such as [1].
2. In Table 1, the proposed method does not demonstrate significant improvements on certain datasets, particularly on the CIFAR-10 dataset. Regarding this phenomenon, the authors need to provide a detailed analysis.
3. In the treatment of dynamic information integration in MHSA (Eqn. (6)), it is noted that not all datasets contain annotated foregrounds. The necessity for manual foreground annotation presents a significant challenge to the practical implementation of this method.
4. minor issues: \
   a.  This paper lacks a detailed introduction to Transformers. The authors should provide a conceptual overview and offer clearer explanations of certain variables in the equations. For instance, the meanings of $(a_{ij})$ in Eqn. (2) and $(z_{im})$ in Eqn. (4) should be explicitly stated.

   b. If Eqn. (4) is correct, the symbol labeled as $(X)$ in Fig. 1 should be $(Y)$.
5. In Eqn. (8), the authors constrain the static information integration in FFN by specifying aggregation rules. However, it is worth considering whether the number of samples used to establish these rules could also impact the final results.
6. Lack of explanation why the performance improvements under other computational forms are not as significant in Table 2.


[1] Trockman, Asher, and J. Zico Kolter. "Mimetic initialization of self-attention layers." International Conference on Machine Learning. PMLR, 2023.

**Questions:**

See weakness

**Limitations:**

The paper has already discussed its limitations and potential impacts.

---

> ### Author Rebuttal · Authors · 2024-08-06
>
> Thank you for your review and comments. We are pleased that you find our work combining Transformers with biological vision error mechanisms to be novel and that you find our methods and findings insightful and inspiring. Below are our responses to each of your comments (each of your comments is highlighted in italics).
>
> > Q1: *The visualization from vertical lines to diagonals in the attention matrices presented in Fig. 2(a,c) lacks sufficient clarity. This visualization is essential for validating the authors' information integration hypothesis in MHSA. It is recommended that the authors refer to similar visualization results in studies such as [1].*
> >
>
> Thank you for pointing this out. The lack of clarity in Figures 2(a) and 2(c) is due to compression artifacts in the PDF version of the paper, which diminished the visibility of the diagonal and vertical lines. As suggested, similar observations can be found in the study you referenced [1]. Additionally, similar visualizations are presented in [2], but their focus is on parameter initialization or architectural design, which differs from our study on Transformer error mechanisms.
>
> [1] Trockman, Asher, and J. Zico Kolter. "Mimetic initialization of self-attention layers." International Conference on Machine Learning. PMLR, 2023.
>
> [2] Chang, Shuning, et al. "Making vision transformers efficient from a token sparsification view." *Proceedings of the IEEE/CVF Conference on Computer Vision and Pattern Recognition*. 2023.
>
> > Q2: *In Table 1, the proposed method does not demonstrate significant improvements on certain datasets, particularly on the CIFAR-10 dataset. Regarding this phenomenon, the authors need to provide a detailed analysis.*
> >
>
> Indeed. It is important to note that Transformer Doctor shows less improvement on smaller datasets such as CIFAR-10 compared to larger datasets. There are two potential reasons for this phenomenon. First, for smaller datasets, errors in Transformer recognition are not solely due to conjunction errors; they may also arise from insufficient feature extraction at early network stages (lines 283-289 of the paper). Second, during the treatment of Transformers, the small image size of CIFAR-10 limits the effectiveness of dynamic aggregation constraints. Consequently, we only employed static aggregation constraints for treatment. In contrast, larger datasets benefit from both dynamic and static aggregation constraints, resulting in better performance. More detailed results are provided in Table 3 of Appendix G.
>
> > Q3: *In the treatment of dynamic information integration in MHSA (Eqn. (6)), it is noted that not all datasets contain annotated foregrounds. The necessity for manual foreground annotation presents a significant challenge to the practical implementation of this method.*
> >
>
> While we acknowledge that not all datasets come with annotated foregrounds, it is important to note that the amount of foreground annotation required for dynamic information integration treatment is quite minimal. For instance, in our experiments using ImageNet-S, the number of images with foreground annotations in the training set is 9,190 out of a total of 1,200,000 images. Despite this small proportion, excellent results were achieved.
>
> Moreover, our further experiments indicate that selectively annotating low-confidence images in the training set is more beneficial compared to randomly annotating some images. Therefore, in practical applications, one can selectively annotate a very small number of low-confidence images (e.g., 5-10 per class) in the training set to perform the Transformer treatment effectively.
>
> > Q4: *This paper lacks a detailed introduction to Transformers. The authors should provide a conceptual overview and offer clearer explanations of certain variables in the equations. For instance, the meanings of ($a_{ij}$) in Eqn.(2) and ($z_{im}$) in Eqn.(4) should be explicitly stated.*
> >
>
> Thank you for your suggestion. We provide an explanation of $\mathbf{a}$ on line 129 of the paper, where $\mathbf{a}\_{ij}$ denotes the weight associated with the $j$-th token when integrating information to form the $i$-th token in inter-token information integration. Additionally, $\mathbf{z}$ is explained on line 154 of the paper, where $\mathbf{z}\_{im}$ represents the weight corresponding to the $m$-th dimension when integrating information to form the $i$-th token in intra-token information integration. We will include these clarifications in the Methods section of the revised manuscript.
>
> > Q5: *If Eqn.(4) is correct, the symbol labeled as $X$ in Fig.1 should be $Y$.*
> >
>
> We apologize for this error. We have corrected the label and thoroughly reviewed the entire manuscript to ensure that all symbols are accurate.
>
> > Q6: *Lack of explanation why the performance improvements under other computational forms are not as significant in Table 2.*
> >
>
> Thank you for your question. In Table 2, the computational forms that show the best performance improvements all involve gradient-based methods. As described on lines 223-225 of the paper, gradients help differentiate the importance of each head in MHSA, leading to more accurate multi-head integration weights $\hat{a}$. Other computational methods, such as using the minimum, maximum, or average across all heads, result in less accurate multi-head integration weights $a^h$ because they include weights from less important heads. Consequently, these methods do not achieve as good results when used for dynamic integration constraints.
>
> Similarly, as noted on lines 238-240 of the paper, gradients establish a connection between integration weights $z$ and specific classes. This results in more precise integration weights $\hat{z}$ for each class and avoids the issue of constraining redundant dimensions during static integration constraints. Thus, gradient-based methods yield better performance improvements. We will include this explanation in the revised version of the paper.

---

> ### Comment · Reviewer_3nqh · 2024-08-13
>
> Thank you for your reply, my concerns have been resolved. So I vote for acceptance.

---

> > ### Author Response · Authors · 2024-08-13
> >
> > Thank you for your valuable review and suggestions. We are pleased to hear that your concerns have been addressed.

---

### Official Review · Reviewer_BM6N · 2024-07-12

**Soundness:** 2
**Presentation:** 2
**Contribution:** 2
**Rating:** 5
**Confidence:** 3

**Summary:**

This study introduced a framework, namely Transformer Doctor, to reduce internal errors, e.g. conjunction errors, in a general vision transformer model. Building upon the information integration hypothesis, the proposed method performs several constraints, including heuristic dynamic constraints and rule-driven static constraints, to enhance information integration at higher layers. The experiential results are conducted on five classification datasets using seven small-scale vision transformer models.

**Strengths:**

-- This study investigates the topic of improving vision transformers, which seems interesting and of practical importance.

-- The proposed method is inspired by solutions in biological vision. The solution might be reasonable.

**Weaknesses:**

-- This is a pure empirical paper with no theoretical results. However, the experimental results are insufficient to fully convince the reviewer.
* The vanilla transformer results reported in Table 1 do not align with their original paper. e.g. the DeiT-Tiny is reported at 72.2% on ImageNet, which is even higher than the improved results (66.8% $\to$ 70.6%).
* This study is only evaluated on small-scale vision transformers, with no results presented for large-scale versions.
* The evaluation is only conducted on 10 samples per class (Appendix F). The sample size seems insufficient.
* Table 1 can be further improved by presenting results with both the vanilla and blank training.

-- The reviewer is unclear about the potential mechanism/logic/reason of the proposed method on the following questions
* why the proposed method can reduce the conjunction error
* To what extent (fully/partially) can the proposed reduce the error
* Besides the information integration hypothesis, do you rely on other assumptions?
* Is the information integration hypothesis valid in vision transformers, especially for large-scale versions?

The above questions might be potentially solved by adding theoretical analysis or more detailed explanations.

-- This paper has many typos, even in the result presentation (e.g. Table 1 BeiT/Imagenet-1K cell).

**Questions:**

See above weakness.

**Limitations:**

As I can see, this paper has no potential negative societal impact.

---

> ### Author Rebuttal · Authors · 2024-08-06
>
> Thank you for your diligent review and comments. We are pleased that you find this research on improving Vision Transformer both interesting and practically valuable. We are also glad that you consider our biologically inspired approach to be reasonable. Below are our responses to each of your comments (each of your comments is highlighted in italics).
>
> > Q1: *The vanilla transformer results reported in Table 1 do not align with their original paper. e.g. the DeiT-Tiny is reported at 72.2% on ImageNet, which is even higher than the improved results (66.8% → 70.6%).*
> >
>
> We apologize for any confusion caused. To achieve the best possible performance for models such as ViT and TNT within our limited computational resources, we adhered to the following detailed experimental settings:
>
> We used data augmentation methods including Auto Contrast, Equalize, Invert, Rotate, Posterize Increasing, Solarize Increasing, and Solarize Add. The AdamW optimizer was employed with a momentum setting of 0.9, weight decay of 0.05, and epsilon of 1e-8. The learning rate scheduler used was CosineLRScheduler with an initial learning rate of 1e-2, a minimum learning rate of 1e-5, no warmup steps, a cycle limit of 1, and T_max set to 300. The training batch size was 256, and the models were trained for 300 epochs.
>
> These settings differ from the commonly used configurations for DeiT, such as a total batch size of 1024 and an initial learning rate of 1e-3. However, it is important to emphasize that this does not impact the fairness of exploring the internal error mechanisms of the models or addressing their errors. During the experimental phase, the comparison settings in the paper are sufficiently fair; except for the number of training epochs for the treated models being fewer, the experimental settings for the treated and baseline models are identical. The experimental results demonstrate that the proposed method has substantial potential for correcting model errors and enhancing performance (as shown in Table 3 in the appendix). We understand your concerns and will add a more detailed description and further experiments in the paper to clarify this matter. We sincerely hope this response addresses your concerns.
>
> > Q2: *This study is only evaluated on small-scale vision transformers, with no results presented for large-scale versions.*
> >
>
> Thank you for your comment. Transformer Doctor is indeed applicable to large-scale Vision Transformers. To demonstrate this, we have additionally conducted preliminary experiments using ViT-Large on ImageNet-10. The results are as follows:
>
> Base: 61.30; +Blank: 61.50; +IDI:62.80; +ISI:62.50; +IDI, +ISI: 62.90;
>
> These results indicate that Transformer Doctor remains effective for large-scale ViTs. The reason is that for any Vision Transformer architecture, such as ViT, the fundamental structure is similar across different sizes, with variations primarily in the number of blocks, heads, and dimensionality of hidden features. Each block consists of MHSA and FFN components. The information integration hypothesis upon which Transformer Doctor is based comprehensively covers both MHSA and FFN (Sections 3.1 and 3.2). Thus, the size of the model does not affect the fundamental efficacy of Transformer Doctor in enhancing model performance.
>
> > Q3: *The evaluation is only conducted on 10 samples per class (Appendix F). The sample size seems insufficient.*
> >
>
> Thank you for your comment. In fact, the amount of data used for evaluation is more than sufficient and far exceeds 10 samples per class.  The "10 samples per class" mentioned refers not to the evaluation data but to the dynamic information integration constraints during training. This indicates that our proposed dynamic integration method requires only a very small amount of annotated foreground masks per class to achieve good results.
>
> Of course, having more foreground annotations during training can provide more information for dynamic integration constraints, thereby further improving model performance. However, considering practical applications, we opted to use a minimal amount of foreground annotations in our experiments to demonstrate the practicality of our proposed method.
>
> > Q4: *Table 1 can be further improved by presenting results with both the vanilla and blank training.*
> >
>
> Thank you for your valuable suggestion. Due to space constraints in the main text, detailed results for both vanilla and blank training have been presented in Table 3 of Appendix G in our original manuscript. As observed, blank training shows almost no improvement over vanilla training, indicating that the performance gains are largely attributed to the proposed method. We will make an effort to include these results in Table 1 of the main text if space permits.

---

> ### Author Response · Authors · 2024-08-06
> **Rebuttal by Authors [Q5-Q9]**
>
> > Q5: *Why the proposed method can reduce the conjunction error*
> >
>
> Thanks. Taking the conjunction error in MHSA as an example, in the advanced stages of Transformer’s MHSA, the integration weights $\mathbf{a}$ dynamically integrate specific foreground information between tokens for high-confidence samples, while they incorrectly integrate background-related information for low-confidence samples (lines 175-182 of the paper). During the treatment phase, our heuristic dynamic information integration method constrains the integration weights $\mathbf{a}$ through the loss function to encourage integration of foreground token information (lines 231-234 of the paper).
>
> Importantly, we introduced gradients to differentiate the importance of each head in MHSA, obtaining the integration weights $\hat{\mathbf{a}}$ in the multi-head scenario and constraining them (lines 223-227 of the paper). We then update the MHSA parameters through backpropagation of the loss function’s gradient, which helps MHSA produce more accurate integration weights $\mathbf{a}$ and reduces the occurrence of erroneous background information integration.
>
> Similarly, for the conjunction errors in the inter-token information static integration within FFN, we apply constraints to the integration weights $\mathbf{z}$ using the loss function, and update FFN parameters through gradient backpropagation. This process helps FFN generate more accurate integration weights $\mathbf{z}$ and reduces the occurrence of conjunction errors (lines 244-247 of the paper).
>
> We hope these explanations address your concerns.
>
> > Q6: *To what extent (fully/partially) can the proposed reduce the error*
> >
>
> This is an insightful question. The extent to which conjunction errors are reduced can be assessed through the decrease in the loss functions mentioned, specifically $\mathcal{L}\_{IDI}$ in Equation (6) and $\mathcal{L}\_{ISI}$ in Equation (9). In the experiments presented in the paper, both loss functions show a significant downward trend and ultimately converge, indicating that the conjunction errors are reduced to the maximum extent possible based on these loss functions.
>
> However, depending on the dataset size and the choice of various hyperparameters, the loss functions do not decrease to zero but rather stabilize at certain values. Therefore, while conjunction errors are reduced significantly, they are not entirely eliminated. We appreciate your question and will update this discussion in the paper to clarify this aspect further.
>
> > Q7: *Besides the information integration hypothesis, do you rely on other assumptions?*
> >
>
> Thank you for this insightful question. Yes, we do rely on additional assumptions. Specifically, the proposed Transformer Doctor is fundamentally based on the Information Integration Hypothesis. This hypothesis is inspired by conjunction errors observed in biological visual systems[1].
>
> Studies such as [2] suggest that these conjunction errors can be mitigated through certain stimuli and cues. Similarly, during the treatment phase, we assume that such errors within the Transformer can also be improved using specific cues.
>
> In summary, as we mentioned in Section 1, the motivation for this work stems from the error mechanisms observed in biological vision. Thus, Transformer Doctor is closely tied to insights from these related works.
>
> [1] Treisman, Anne M., and Garry Gelade. "A feature-integration theory of attention." *Cognitive Psychology* 12.1 (1980): 97-136.
>
> [2] Prinzmetal, William, David E. Presti, and Michael I. Posner. "Does attention affect visual feature integration?" *Journal of Experimental Psychology: Human Perception and Performance* 12.3 (1986): 361.
>
> > Q8: *Is the information integration hypothesis valid in vision transformers, especially for large-scale versions?*
> >
>
> Thank you for your question. The Information Integration Hypothesis is indeed valid for large-scale Vision Transformers. As addressed in response to your second comment (Q2), the depth of the Transformer, the number of heads, and the dimensionality of features do not affect the information integration process in MHSA and FFN. Quantitative experiments also confirm that the hypothesis is effective for large-scale Vision Transformers, such as ViT-Large, demonstrating similar beneficial results.
>
> > Q9: *This paper has many typos, even in the result presentation (e.g. Table 1 BeiT/Imagenet-1K cell).*
> >
>
> We sincerely apologize for the typographical errors. The issue you mentioned has been addressed in the revised manuscript, and we have conducted a thorough review of the entire paper to avoid similar issues.

---

> ### Comment · Reviewer_BM6N · 2024-08-11
>
> The authors have addressed some of my concerns about experimental results. I have increased my rating.

---

> > ### Author Response · Authors · 2024-08-11
> >
> > Thank you for your positive feedback and recognition of our work. We are pleased that our response addressed your concerns. We sincerely appreciate your valuable comments, which have helped improve this work.

---

### Official Review · Reviewer_ZXFK · 2024-07-13

**Soundness:** 4
**Presentation:** 3
**Contribution:** 3
**Rating:** 7
**Confidence:** 5

**Summary:**

This paper presents Transformer Doctor, which diagnoses the issues with the Transformer attention mechanism and resolves them via several information integration hypotheses. The primary motive of the paper is to identify the source of incorrect information aggregation, which leads to erroneous predictions, and the inspiration comes from biological groundings. The paper proposed Inter-token Information Static Integration, Intra-token Information Static Integration, Heuristic Information Dynamic Integration Therapy, Rule-based Information Static Integration Therapy, Joint Therapy of Dynamic and Static Integration, and extensive evaluation. The paper improves the performance of the Transformer mechanism significantly.

**Strengths:**

1. The paper is well written.
2. Each proposed component is well-motivated and discussed.
3. The step-by-step introduction of the hypothesis from section 3 to section 5 provides a clear picture.
4. The inter-token and intra-token diagnosis hypotheses are very promising for understanding the internals of MSHA.
5. The improved attention maps show the potential of the proposed information integration mechanism.
6. The integration equations are straightforward and easier to understand.

**Weaknesses:**

1. It appears from sec6.1 that the proposed approach requires a pre-trained model. Please correct me if not because I could not find explicit training settings in the paper. If it is trained for longer epochs, it increases the training time.
2. Since the model is pre-trained, even though attention maps are improving, it is unclear whether the gains came from further fine-tuning or information integration. In other words, how much the integration mechanism contributed to the gains?
3. In Figure 4, the regions in red always lie on the object, i.e. without and with transformer doctor applied. How can it be inferred that the attended region after using the doctor mechanism is the sole reason for improved performance? Similarly Fig12.
4. Selecting the gradient based on the actual class label is possible with supervised learning. Hence, how can this method be applied to DINO-like self-supervised methods, given that DINO also uses a Transformer?

Note: I am open to adjusting ratings if my concerns are resolved, especially 1,2,3.

**Questions:**

See weakness.

**Limitations:**

Overall, the main limitations are the use of pre-trained models as a baseline, unclear hypothesis verification on the improved attention maps, and usability across self-supervised models.

---

> ### Author Rebuttal · Authors · 2024-08-06
>
> Thank you for your diligent review and comments. We are pleased that you found the paper well-written and appreciated our step-by-step introduction of the proposed methods and hypotheses. Your recognition of our diagnostic hypotheses and the potential of the proposed information integration mechanism is highly encouraging. Below are our responses to each of your comments (each of your comments is highlighted in italics).
>
> > Q1: *It appears from sec6.1 that the proposed approach requires a pre-trained model. Please correct me if not because I could not find explicit training settings in the paper. If it is trained for longer epochs, it increases the training time.*
> >
>
> We apologize for any confusion caused. The proposed Transformer Doctor indeed diagnoses and treats an already trained model, i.e., a model that has undergone normal training and fitting, as mentioned on lines 261-263 of the paper. Similar to most interpretability methods based on trained models, the proposed information integration hypothesis assumes that the trained model exhibits error mechanisms akin to those in biological vision (lines 126-129 of the paper). Specifically, in the diagnostic and therapeutic experimental setup, the pre-trained model was trained for 300 epochs, and continuing training does not improve performance (Table 3 in Appendix G). During the diagnosis and treatment phase, we retrain for an additional 100 epochs using the same training settings, which yields excellent results without significantly increasing the overall training time. We will emphasize this part in the paper to avoid any further confusion.
>
> > Q2: *Since the model is pre-trained, even though attention maps are improving, it is unclear whether the gains came from further fine-tuning or information integration. In other words, how much the integration mechanism contributed to the gains?*
> >
>
> Thank you for your comment. The pre-trained model used in our study is a fully trained and fitted model, so further fine-tuning does not enhance its performance. However, significant improvements in accuracy are observed only after diagnosis and treatment with Transformer Doctor. This can be seen in Table 3 of Appendix G. For instance, in the case of ViT-Tiny on ImageNet-10, the baseline accuracy is 78.90%. After further fine-tuning ("*+Blank*"), the accuracy remains almost unchanged. However, with the application of information dynamic integration constraints ("*+IDI*"), information static integration constraints ("*+ISI*"), and both dynamic and static integration constraints ("*+IDI, ISI*"), the accuracy improves by 1.40%, 1.60%, and 2.00% respectively. Therefore, the performance gains are primarily due to the integration mechanism improvements rather than further fine-tuning. Similarly, the improvements in attention maps are also mainly attributable to the integration mechanism rather than further fine-tuning.
>
> > Q3: *In Figure 4, the regions in red always lie on the object, i.e. without and with transformer doctor applied. How can it be inferred that the attended region after using the doctor mechanism is the sole reason for improved performance? Similarly Fig12.*
> >
>
> Thank you for your valuable comments. It is true that in Figures 4 and 12, some images show red regions on the object both before and after applying Transformer Doctor. However, after applying Transformer Doctor, the red regions on some objects become more focused (e.g., the "orange" in Figure 12), indicating that the model is attending to more useful features. Additionally, in some images, the red regions initially lie on the background, but after applying Transformer Doctor, they shift to the object (e.g., the "leopard" in Figure 4), suggesting that the model is now focusing on information most relevant to the prediction task. Furthermore, referring back to Table 3 in Appendix G, the performance improvement does not come from further fine-tuning, indicating that it is solely due to the proposed Transformer Doctor mechanism. We hope these explanations address your concerns.
>
> > Q4: *Selecting the gradient based on the actual class label is possible with supervised learning. Hence, how can this method be applied to DINO-like self-supervised methods, given that DINO also uses a Transformer?*
> >
>
> This is an excellent question. While our method has been validated on most commonly used Transformer architectures, it is indeed challenging to directly apply the proposed method to self-supervised learning methods. As you correctly pointed out, the calculation of gradients in our method requires the availability of actual labels. In methods like DINO, which update parameters through contrastive learning and momentum updates for visual representation learning, it is difficult to compute gradients related to specific true labels, significantly limiting the applicability of our method.
>
> A potential approach is to utilize the intrinsic mechanisms within self-supervised learning methods to find alternatives to the gradient-based approach. For instance, in DINO, the contrastive learning signal between the teacher and student networks can be used to estimate the importance of different heads in the Transformer. This signal could serve as a proxy for the gradients in our method.
>
> Thank you for raising this important question again. Exploring the integration hypothesis and diagnostic and treatment methods for more complex tasks like self-supervised learning will be a direction of our ongoing research.

---

> ### Comment · Reviewer_ZXFK · 2024-08-09
> **Response to Authors**
>
> I thank the authors for the detailed responses to my comments.
>
> Overall, I am convinced about my question on the source of the accuracy gains i.e. higher training epochs or information integration.
>
> I have read comments from other reviewers and responses to them and they are in line with my understanding of the paper.
>
> I have a particular concern about the reported results for DieT commented on by reviewer BM6N. I agree with the reviewer that resource constraints lead to smaller batch sizes for training and thus unfair comparison. However, I also applaud the authors for training the baseline with the same baseline settings. At the same time, I suggest authors to match the exact settings of the baselines because the results presented in this paper would become a baseline for comparison in future. Hence to avoid confusion among readers, it is advisable to add a detailed training setting section in the supplementary and also a small footnote or caption in the table stating the reasons for the accuracy difference of DieT while also adding the results from the original paper.
>
> Based on the reviews, I vote for acceptance. I have increased my rating.

---

> > ### Author Response · Authors · 2024-08-09
> >
> > Thank you very much for your thoughtful review and the positive feedback. We sincerely appreciate your acknowledgment of our efforts to address your concerns and your recognition of the improvements made in the paper.
> >
> > We understand your concern about the reported results for DieT, as highlighted by reviewer BM6N. We agree that ensuring fair comparisons is critical. To address this, we will include a detailed training settings section in the supplementary materials and provide a footnote or caption in the relevant tables to explain the accuracy differences for DieT. We will also add the results from the original paper for completeness.
> >
> > Once again, we sincerely thank you for your valuable feedback and for recommending our work for acceptance. Your insights have been instrumental in improving the quality of our paper.

---

### Author Rebuttal · Authors · 2024-08-06

Dear Reviewers ZXFK, BM6N, 3nqh, and KQuQ,

Thank you for your diligent reviews and constructive feedback. We particularly appreciate your recognition of the novelty and insightfulness of our work and are pleased that you find our approach of integrating error mechanisms from biological vision with Transformers both reasonable and inspiring.

Through our detailed responses to each of your comments, we believe the paper has significantly improved. We have addressed each comment individually and collected the following common concerns raised by the reviewers. We hope our responses address your concerns and would be grateful if you could consider raising your scores. If you have any additional questions, we are more than happy to engage in further discussion.

> Reviewer BM6N pointed out that the number of foreground annotations used for evaluation is insufficient, while Reviewer 3nqh noted that not all datasets have foreground annotations, which limits further usage.
>

We apologize for the confusion caused to both reviewers. To address Reviewer BM6N's concern, it is important to clarify that the foreground annotations are used for training, not evaluation. Considering that not all datasets have foreground annotations, our method achieves good performance with only a small number of foreground annotations. Regarding Reviewer 3nqh's concern, we emphasize that during the dynamic information integration treatment, very few manually annotated foregrounds are required to achieve significant improvements, which does not hinder the practical applicability of our method. For more details, please refer to our responses to each of your individual comments.

> Reviewer BM6N raised concerns about training baselines and the validation of large-scale Transformers.
>

We fully understand the reviewer's concerns. In response to these issues, we have provided detailed explanations and added relevant experiments below the reviewer's comments. It is important to emphasize that the experiments in the paper were conducted under fair experimental settings for comparison and analysis. We sincerely hope that the corresponding responses address the reviewer's concerns effectively.

> Reviewers ZXFK and 3nqh both raised issues related to gradients. One mentioned the applicability of gradients in self-supervised learning methods like DINO, while the other questioned why gradient-based methods are more effective.
>

We apologize for any confusion caused. In the responses to the reviewers' comments, we have provided detailed explanations on the role of gradients, their applicability in DINO, and potential alternatives to gradients. We believe these responses address the reviewers' concerns.

> Reviewers BM6N and 3nqh both pointed out several typos in the paper.
>

We appreciate the reviewers’ attention to detail in identifying these typos. We have corrected these errors and thoroughly reviewed the entire paper.

In addition to addressing the above issues, detailed responses to each reviewer’s comments can be found below their respective feedback. Thank you once again to all the reviewers for their meticulous review and valuable comments, which have significantly improved the paper.

---

### Decision · Program_Chairs · 2024-09-25

**Decision:**

Accept (poster)

**Comment:**

The reviewers unanimously support the paper. A couple of reviewers are concerns with DeiT evaluation settings and fairness in comparison. This concern still remain partly unaddressed.

Overall, the paper looks valid and it has its strength, despite the remaining concern.